# Mask Matching Transformer for Few-Shot Segmentation

**Siyu Jiao**[1,2]*,   **Gengwei Zhang**[3],   **Shant Navasardyan**[4],   **Ling Chen**[3],   **Yao Zhao**[1,2],
**Yunchao Wei**[1,2],   **Humphrey Shi**[4]

[1] Institute of Information Science, Beijing Jiaotong University
[2] Beijing Key Laboratory of Advanced Information Science and Network
[3] AAII, University of Technology Sydney    [4] Picsart AI Research (PAIR)
jiaosiyu@bjtu.edu.cn

## Abstract

In this paper, we aim to tackle the challenging few-shot segmentation task from a new perspective. Typical methods follow the paradigm to firstly learn prototypical features from support images and then match query features in pixel-level to obtain segmentation results. However, to obtain satisfactory segments, such a paradigm needs to couple the learning of the matching operations with heavy segmentation modules, limiting the flexibility of design and increasing the learning complexity. To alleviate this issue, we propose Mask Matching Transformer (MM-Former), a new paradigm for the few-shot segmentation task. Specifically, MM-Former first uses a class-agnostic segmenter to decompose the query image into multiple segment proposals. Then, a simple matching mechanism is applied to merge the related segment proposals into the final mask guided by the support images. The advantages of our MM-Former are two-fold. First, the MM-Former follows the paradigm of *decompose first and then blend*, allowing our method to benefit from the advanced potential objects segmenter to produce high-quality mask proposals for query images. Second, the mission of prototypical features is relaxed to learn coefficients to fuse correct ones within a proposal pool, making the MM-Former be well generalized to complex scenarios or cases. We conduct extensive experiments on the popular COCO-$20^i$ and Pascal-$5^i$ benchmarks. Competitive results well demonstrate the effectiveness and the generalization ability of our MM-Former. Code is available at github.com/Picsart-AI-Research/Mask-Matching-Transformer.

## 1   Introduction

Semantic segmentation, one of the fundamental tasks in computer vision, has achieved a grand success [3, 5, 37, 12] in recent years with the advantages of deep learning techniques [11] and large-scale annotated datasets [14, 6]. However, the presence of data samples naturally abides by a long-tailed distribution where the overwhelming majority of categories have very few samples. Therefore, few-shot segmentation [21, 27, 35] is introduced to segment objects of the tail categories only according to a minimal number of labels.

Mainstream few-shot segmentation approaches typically follow the learning-to-learning fashion, where a network is trained with episodic training to segment objects conditioned on a handful of labeled samples. The fundamental idea behind it is how to effectively use the information provided by the labeled samples (called *support*) to segment the test (referred to as the *query*) image. Early

---

*Work done during an internship at Picsart AI Research (PAIR).

36th Conference on Neural Information Processing Systems (NeurIPS 2022).

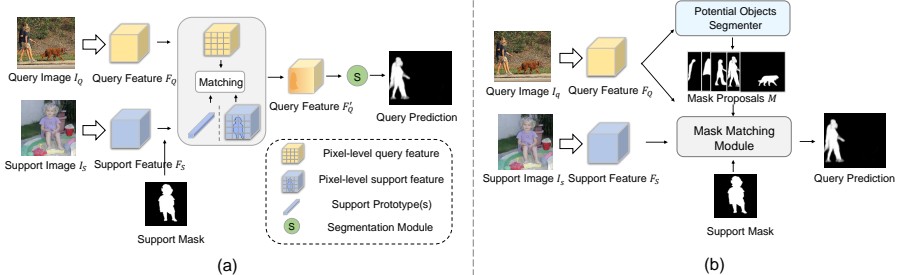

Figure 1: Comparison between the existing few-shot segmentation framework and our MM-Former. (a) Previous works first match support prototype (s) [33] or pixel-level support feature [34, 32] with the pixel-level query feature, then pass the matched feature through the segmentation module to obtain the query prediction. (b) Our MM-Former decouples the segmentation and the matching problems for the few-shot segmentation task. The query segmentation result is acquired with a simple Mask Matching module, which operates on the support samples and a set of query mask proposals.

works [27, 36, 33, 24, 30] achieve this by first extracting one or few semantic-level prototypes from features of support images, and then pixels in the query feature map are matched (activated) by the support prototypes to obtain the segmentation results. We refer to this kind of method as "few-to-many" matching paradigm since the number of support prototypes is typically much less (*e.g.,* two to three orders of magnitude less) than the number of pixels in the query feature map. While acceptable results were obtained, this few-to-many matching paradigm turns out to be restricted in segmentation performance due to the information loss in extracting prototypes. Therefore recent advances [34, 26, 19] proceed to a "many-to-many" matching fashion. Concretely, pixel-level relationships are modeled between the support and query feature maps either by attention machanism [34, 26] or 4D convolutions [19]. Benefiting from these advanced techniques, the many-to-many matching approaches exhibit excellent performance over the few-to-many matching counterparts. **Overall**, the aforementioned approaches construct modules combining the matching operation with segmentation modules and optimizing them jointly. For the sake of improving the segmentation quality, techniques of context modeling module such as atrous spatial pooling pyramid [3], self-attention [39] or multi-scale feature fusion [24] are integrated with the matching operations [33] and then are simultaneously learned via the episodic training [30, 34, 24]. However, this joint learning fashion not only vastly increases the learning complexity, but also makes it hard to distinguish the effect of matching modules in few-shot segmentation.

Therefore, in this work, we steer toward a different perspective for few-shot segmentation: decoupling the learning of segmentation and matching modules as illustrated in Fig. 1. Rather than being matched with the pixel-level query features, the support samples are directly matched with a few class-agnostic query mask proposals, forming a "few-to-few" matching paradigm. By performing matching in the mask level, several advantages are provided: 1) Such a few-to-few matching paradigm releases matching from the segmentation module and focuses on the matching problem itself. 2) It reduces the training complexity, thus a simple few-to-few matching is enough for solving the few-shot segmentation problem. 3) While previous works turned out to be overfitting when using high-level features for matching and predicting the segmentation mask [24, 27, 34], the learning of our matching and segmentation module would not affect each other and hence avoids this daunting problem.

To achieve this few-to-few matching paradigm, we introduce a two-stage framework, named Mask Matching Transformer (dubbed as MM-Former), that generates mask proposals for the query image in the first stage and then matches the support samples with the mask proposals in the second stage. Recently, MaskFormer [4, 5] formulates semantic segmentation as a mask classification problem, which obtains semantic segmentation results by combining the predictions of binary masks and the corresponding classification scores, where the masks and the scores are both obtained by using a transformer decoder. It provides the flexibility for segmenting an image of high quality without knowing the categories of the objects in advance. Inspired by this, we also use the same transformer decoder as in [4] to predict a set of class-agnostic masks based on the query image only. To further determine the target objects indicated by the support annotation, a simple Mask Matching Module is constructed. Given the features extracted from both support and query samples, the Mask Matching Module obtains prototypes from both support and query features through masked global average pooling [27]. Further, a matching operation is applied to match the supports with all query proposals and produces a set of coefficients for each query candidate. The final segmentation result for a given

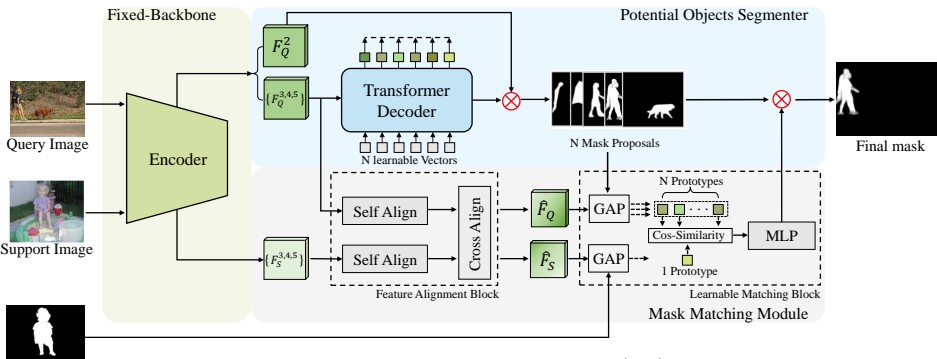

**Figure 2: An overview of the proposed MM-Former.** The support and query images are passed through an weight-shared encoder. The support and query features from the Encoder donate as $\mathbf{F}_S$ and $\mathbf{F}_Q$, respectively. We first train a **Potential Objects Segmenter**, which can predict all the proposal objects in one image (color in blue). Then we use $F_S$ in stage two as guidance information to collaboratively mine the final segmentation by **Mask Matching Module** (color in grey).

support(s)-query pair is acquired by combining the mask proposals according to the coefficients. In addition, to resolve the problem of representation misalignment caused by the differences between query and support images, a Feature Alignment Block is integrated into the Mask-Matching Module. Concretely, since the features for all images are extracted with a fixed network, they may not be aligned well in the feature space, especially for testing images with novel classes. A Self Alignment block and a Cross Alignment block are introduced to consist of the Feature Alignment Block to align the query and support samples in the feature space so that the matching operation can be safely applied to the extracted features.

We evaluate our MM-Former on two commonly used few-shot segmentation benchmarks: COCO-$20^i$ and Pascal-$5^i$. Our model stands out from previous works on the challenging COCO-$20^i$ dataset. While our MM-Former only performs comparably with previous state-of-the-art methods due to the limited scale of the Pascal dataset, our MM-Former exhibits a strong transferable ability across different datasets (*i.e.*, COCO-$20^i \rightarrow$ Pascal-$5^i$), owing to our superior mask-matching design. **In a nutshell**, our contributions can be summarized as follows: (1) We put forward a new perspective for few-shot segmentation, which decouples the learning of matching and segmentation modules, allowing more flexibility and lower training complexity. (2) We introduce a simple two-stage framework named MM-Former that efficiently matches the support samples with a set of query mask proposals to obtain segmentation results. (3) Extensive evaluations on COCO-$20^i$ and Pascal-$5^i$ demonstrate the potential of the method to be a robust baseline in the few-to-few matching paradigm.

## 2 Methodology

**Problem Setting**: Few-shot segmentation aims at training a segmentation model that can segment novel objects with very few labeled samples. Specifically, given two image sets $D_{train}$ and $D_{test}$ with category set $C_{train}$ and $C_{test}$ respectively, where $C_{train}$ and $C_{test}$ are disjoint in terms of object categories ($C_{train} \cap C_{test} = \emptyset$). The model trained on $D_{train}$ is directly applied to test on $D_{test}$. The episodic paradigm was adopted in [24, 38] to train and evaluate few-shot models. A $k$-shot episode $\{\{I_s\}^k, I_q\}$ is composed of $k$ support images $I_s$ and a query image $I_q$, all $\{I_s\}^k$ and $I_q$ contain objects from the same category. We estimate the number of episodes for training and testing set are $N_{train}$ and $N_{test}$, the training set and test set can be represented by $D_{train} = \{\{I_s\}^k, I_q\}^{N_{train}}$ and $D_{test} = \{\{I_s\}^k, I_q\}^{N_{test}}$. Note that both support masks $M_s$ and query masks $M_q$ are available for training, and only support masks $M_s$ are accessible during testing.

**Overview**: The proposed architecture can be divided into three parts, *i.e.*, Backbone Network, Potential Objects Segmenter and Mask Matching Module. Specifically, the Backbone Network is used to extract features only, whose parameters are fixed during the training. The Potential Objects Segmenter (dubbed as POS) is applied to produce multiple mask proposals that may contain potential object regions within the given image. The Mask Matching Module (dubbed as MM module) takes support cues as guidance to choose the most likely ones from the mask proposals. The selected masks are finally merged into the target output. The complete diagram of the architecture is shown in Fig. 2. Each of the modules will be explained in detail in the following subsections.

## 2.1 Feature Extraction Module

We adopt a ResNet [11] to extract features for input images. Unlike previous few shot segmentation methods [24, 38, 31] using Atrous Convolution to replace strides in convolutions for keeping larger resolutions, we keep the original structure of ResNet following [5]. We use the outputs from the last three layers in the following modules and named them as $F_S$ and $F_Q$ for $I_S$ and $I_Q$, where $F = \left\{ F^i \right\}, i \in [3, 4, 5]$ and $F$ is the features of $I_S$ or $I_Q$, $i$ is the layer index of backbone. We further extract the output of query $layer2$ to obtain the segmentation mask (named as $F_Q^2$). $F^2$, $F^3$, $F^4$ and $F^5$ have strides of $\{4, 8, 16, 32\}$ with respect to the input image.

## 2.2 Potential Objects Segmenter

The POS aims to segment all the objects in one image. Follow Mask2Former [4], a standard transformer decoder [25] is used to compute cross attention between $F_Q$ and N learnable embeddings. The transformer decoder consists of 3 consecutive transformer layers, each of which takes the corresponding $F^i$ as an input. Each layer in the transformer decoder can be formulated as

$$E^{l+1} = \text{TLayer}^l(E^l, F^i), \tag{1}$$

where $E^l$ and $E^{l+1}$ represent the N learnable embeddings before and after applying the transformer layer respectively. TLayer denote a transformer decoder layer. We simplify the representation of transformer decoder, whereas we conduct the same pipeline proposed by Mask2Former. The output of the transformer decoder is multiplied with $F_Q^2$ to get N mask proposals $M \in \mathbb{R}^{N \times H/4 \times W/4}$. Note that Sigmoid is applied to normalize all mask proposals to $[0, 1]$. Besides, our POS abandons the classifier of Mask2Former since we don't need to classify the mask proposals.

## 2.3 Mask Matching (MM) Module

In MM, our goal is to use support cues as guidance to match the relevant masks. The building blocks of MM are a Feature Alignment block and a Learnable Matching block. We first apply Feature Alignment block to align $F_Q$ and $F_S$ from the pixel level. Then, the Learnable Matching block matches appropriate query masks correspondence to the support images.

**Feature Alignment Block**: We achieve the alignment using two types of building blocks: a Self-Alignment block and a Cross-Alignment block. The complete architecture is shown in Fig. 3.

We adopt the Self-Alignment block to align features in each channel. Inspired from Polarized Self-Attention [16], we design a non-parametric block to normalize representations. Specifically, the input feature map $F \in \mathbb{R}^{c \times hw}$ is first averaged at the channel dimension to obtain $F_{avg} \in \mathbb{R}^{1 \times hw}$. $F_{avg}$ is regarded as an anchor to obtain the attention weight $A \in \mathbb{R}^{c \times 1}$ by matrix multiplication: $A = F F_{avg}^T$, which represents the weights of different channels. $A$ is used to activate the

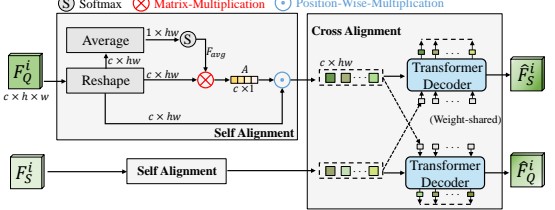

Figure 3: Details of **Feature Alignment Block**. Note *Average* means channel-wise average.

feature by position-wise-multiplication (*i.e.*, expand at the spatial dimension and perform point-wise multiplication with the feature). In this way, the input feature is adjusted across the channel dimension, and the outliers are expected to be smoothed. Note that the Self Alignment block processes $F_S$ and $F_Q$ individually and does not involve interactions across images.

The Cross-Alignment block is introduced to mitigate divergence across different images. $F_S$ and $F_Q$ are fed into two weight-shared transformer decoders in parallel. We take $i^{th}$ layer as an example in Fig. 3, which can be formulated as

$$\hat{F}_Q^i = \text{MLP}(\text{MHAtten}(F_Q^i, F_S^i, F_S^i)),$$
$$\hat{F}_S^i = \text{MLP}(\text{MHAtten}(F_S^i, F_Q^i, F_Q^i)), \tag{2}$$

where $F_Q^i$ and $F_S^i$ represent the $i^{th}$ layer feature in $F_Q$ and $F_S$. $\hat{F}^i$ represents the alignment features. ($\hat{F} = \left\{ \hat{F}^i \right\}_i^L, i \in [3, 4, 5]$). MLP denote MultiLayer Perceptron and MHAtten represents multi-

head attention [25]

$$\text{MHAtten}(q, k, v) = \text{softmax}(\frac{qk^T}{\sqrt{d_k}})v, \tag{3}$$

where $q, k, v$ mean three matrices, and $d_k$ is the dimension of $q$ and $k$ elements. $k, v$ are downsampled to $\frac{1}{32}$ of the original resolution to save computation. To distinguish from the phrase "Query Image" in few-shot segmentation and "matrix Q" in Transformer, we name "Query Image" as **Q** and "matrix Q" as **q**. We simplify the representation of $\text{MHAtten}$ and omit some Shortcut-Connections and Layer Normalizations in transformer decoder, whereas we conduct the same pipeline with the standard transformer.

**Learnable Matching Block**: After acquiring $\hat{F}_S$ and $\hat{F}_Q$, we first apply masked global average pooling (GAP) [38, 27, 24, 31] on each $\hat{F}^i$ and concat them together to generate prototypes for support ground-truth and N query mask proposals. Named as $\{P_S^{gt}\}$, $\{P_Q^i\}$, $P_S^{gt}, P_Q^n \in \mathbb{R}^{3d}$ and $n \in [1, 2...N]$. Here $d$ represents the dimension of $\hat{F}^i$, $3d$ is achieved by concatenating $\{\hat{F}^3, \hat{F}^4, \hat{F}^5\}$ together. We use cosine distance to measure the similarity between the prototypes of $P_S^{gt}$ and $P_Q^n$.

In some cases, the mask corresponding to the prototype with the highest similarity may not be complete (*e.g.*, the support image has only parts of the object). So, we further use an MLP layer to merge corresponding masks. The detailed diagram can be formulated as

$$S = \cos(P_S^{gt}, P_Q^n), n \in [1, 2...N],$$
$$\hat{M} = M \times \text{MLP}(S), S \in R^{1 \times N}, \tag{4}$$

where $\hat{M}$ is our final result, MLP and $\cos$ indicate the fully connected operation and the cosine similarity. We take N similarities ($S$) as the input of MLP, and use the output to perform a weighted average of N mask proposals. Note that we do not select the mask with the highest similarity directly, our ablation studies prove that using this block can help improve the performance.

## 2.4 Objective

In POS, we adopt segmentation loss functions proposed by Mask2Former (denote as $\mathcal{L}_P$). We apply Hungarian algorithm to match mask proposals with groud-truth and only conduct Dice Loss to supervise on masks with the best matching.

In MM module, we conduct Dice Loss on $\hat{M}$ (denote as $\mathcal{L}_M$) and design a contrastive loss to constrain the cross-alignment module. Our goal is to make prototypes for the same class more similar while different classes less similar by constraining $S$. We first normalize $S$ to $[0, 1]$ by min-max normalization $\hat{S} = \frac{S - \min(S)}{\max(S) - \min(S) + \varepsilon}$. Then we calculate IoU between N mask proposals and Query ground-truth. We assume that mask proposals contain various objects in query images. It is unrealistic to constrain the corresponding prototypes across different categories since it is hard to acquire the proper similarity among them. Therefore, we apply a criterion at the location of $\max(\text{IoU})$ and denote the point as *positive* point $\hat{S}_{pos}$. Only constrain on $\hat{S}_{pos}$ may lead all outputs of Cross Alignment block tend to be same. Thus, we add a constraint on the point in $\hat{S}$ corresponding to the lowest IoU, and denote it as *negative* point $\hat{S}_{neg}$. We assign $y_{pos} = 1$ and $y_{neg} = 0$ to $\hat{S}_{pos}$ and $\hat{S}_{neg}$ during the optimization, respectively. Therefore, the cross-alignment loss $\mathcal{L}_{co}$ can be defined as

$$\mathcal{L}_{co} = -\frac{1}{2}(y_{pos} \log \hat{S}_{pos} + (1 - y_{neg}) \log (1 - \hat{S}_{neg})) \tag{5}$$

Thus, the final loss function can be formulated as $\mathcal{L} = \mathcal{L}_P + \lambda_1 \mathcal{L}_M + \lambda_2 \mathcal{L}_{co}$, where $\lambda_1$ and $\lambda_2$ are constants and are set to 10 and 6 in our experiments.

## 2.5 Training Strategy

In order to avoid the mutual influence of POS and MM module during training, we propose a two-stage training strategy of first training POS and then training MM. In addition, by decoupling POS and MM, the network can share the same POS under 1-shot and K-shot settings, which greatly improves the training efficiency.

Table 1: Comparison with state-of-the-art methods on COCO-$20^i$. Best results are shown in bold.

| Method | Backbone | 1-shot | | | | | 5-shot | | | | |
|---|---|---|---|---|---|---|---|---|---|---|---|
| | | $5^0$ | $5^1$ | $5^2$ | $5^3$ | Mean | $5^0$ | $5^1$ | $5^2$ | $5^3$ | Mean |
| PFENet[TPAMI20] [24] | Res-50 | 34.3 | 33.0 | 32.3 | 33.1 | 32.4 | 38.5 | 38.6 | 38.2 | 34.3 | 37.4 |
| SCL[CVPR21] [31] | | 36.4 | 38.6 | 37.5 | 35.4 | 37.0 | 38.9 | 40.5 | 41.5 | 38.7 | 39.9 |
| ASGNet[CVPR21] [13] | | - | - | - | - | 34.6 | - | - | - | - | 42.5 |
| REPRI[CVPR21] [1] | | 31.2 | 38.1 | 33.3 | 33.0 | 34.0 | 38.5 | 46.2 | 40.0 | 43.6 | 42.1 |
| MM-Net[ICCV21] [28] | | 34.9 | 41.0 | 37.8 | 35.2 | 37.2 | 38.5 | 39.6 | 38.4 | 35.5 | 38.0 |
| CWT[ICCV21] [18] | | 32.3 | 36.0 | 31.6 | 31.6 | 32.9 | 40.1 | 43.8 | 39.0 | 42.4 | 41.3 |
| HSNet[ICCV21] [19] | | 36.3 | 43.1 | 38.7 | 38.7 | 39.2 | 43.3 | 51.3 | **48.2** | 45.0 | 46.9 |
| CyCTR[NeurIPS21] [38] | | 38.9 | 43.0 | 39.6 | 39.8 | 40.3 | 41.1 | 48.9 | 45.2 | 47.0 | 45.6 |
| MM-Former (Ours) | Res-50 | **40.5** | **47.7** | **45.2** | **43.3** | **44.2** | **44.0** | **52.4** | 47.4 | **50.0** | **48.4** |
| Oracle | | 66.1 | 74.3 | 64.8 | 70.4 | 68.9 | 66.1 | 74.3 | 64.8 | 70.4 | 68.9 |

Table 2: Comparison with state-of-the-art methods on PASCAL-$5^i$. Best results are shown in bold.

| Method | Backbone | PASCAL → PASCAL | | COCO → PASCAL | |
|---|---|---|---|---|---|
| | | 1 shot | 5 shot | 1 shot | 5 shot |
| PFENet[TPAMI20] [24] | Res-50 | 60.8 | 61.9 | 61.1 | 63.4 |
| SCL[CVPR21] [31] | | 61.8 | 62.9 | - | - |
| REPRI[CVPR21] [1] | | 59.1 | 66.8 | 63.2 | 67.7 |
| HSNet[ICCV21] [19] | | **64.0** | **69.5** | 61.6 | 68.7 |
| CyCTR[NeurIPS21] [38] | | **64.0** | 67.5 | - | - |
| MM-Former (Ours) | Res-50 | 63.3 | 64.9 | **67.7** | **70.4** |
| Oracle | | 82.5 | 82.5 | 85.8 | 85.8 |

**K-shot Setting**: Based on the two stages training strategy, MM-Former can easily extend to the K-shot setting by averaging knowledge from K samples, *i.e.*, $P_S^{gt}$. Note that after pre-trained the POS, MM-Former can be applied to 1-shot/ K-shot tasks with only a very small amount of training.

## 3 Experiments

### 3.1 Dataset and Evaluation Metric

We conduct experiments on two popular few-shot segmentation benchmarks, Pascal-$5^i$ [9] and COCO-$20^i$ [14], to evaluate our method. Pascal-$5^i$ with extra mask annotations SBD [10] consisting of 20 classes are separated into 4 splits. For each split, 15 classes are used for training and 5 classes for testing. COCO-$20^i$ consists of annotated images from 80 classes. We follow the common data split settings in [20, 38, 19] to divide 80 classes evenly into 4 splits, 60 classes for training and test on 20 classes. 1,000 episodes from the testing split are randomly sampled for evaluation. To quantitatively evaluate the performance, we follow common practice [24, 27, 36, 19, 38], and adopt mean intersection-over-union (mIoU) as the evaluation metrics for experiments.

### 3.2 Implementation details

The training process of our MM-Former is divided into two stages. For the **first** stage, we freeze the ImageNet [6] pre-trained backbone. The POS is trained on Pascal-$5^i$ for 20,000 iterations and 60,000 iterations on COCO-$20^i$, respectively. Learning rate is set to $1e^{-4}$, batch size is set to 8. For the **second** stage, we freeze the parameters of the backbone and the POS, and only train the MM module for 10,000/20,000 iterations on Pascal-$5^i$ / COCO-$20^i$, respectively. Learning rate is set to $1e^{-4}$, batch size is set to 4. For both stages, we use AdamW [17] optimizer with a weight decay of $5e^{-2}$. The learning rate is decreased using the poly schedule with a factor of $0.9$. All images are resized and cropped into $480 \times 480$ for training. We also employ random horizontal flipping and random crop techniques for data augmentation. All the experiments are conducted on a single RTX A6000 GPU. The standard ResNet-50 [11] is adopted as the backbone network.

### 3.3 Comparison with State-of-the-art Methods

We compare the proposed approach with state-of-the-art methods [24, 31, 13, 28, 19, 18, 1, 15, 38] on Pascal-$5^i$ and COCO-$20^i$ datasets. The results are shown in Tab. 1 and Tab. 2.

Table 3: **Ablations on Mask Matching Module (Stage-2)**. In (a), the result in first row is obtained by our baseline. * means non-parameters design (Details in Sec. 2.3). We denote cross-alignment supervised with $\mathcal{L}_{co}$ by ✓✓ in the last row (Sec. 2.4). SA, CA, LM represent Self Alignment block, Cross Alignment block and Learnable Matching block, respectively.

(a) Ablation on the components of **Mask Matching Module.**

| Components | | | Pascal-$5^i$ | COCO-$20^i$ |
|---|---|---|---|---|
| SA | CA | LM | mean IoU | mean IoU |
| | | | 42.1 | 22.4 |
| ✓ | | | 46.4 | 23.6 |
| | ✓* | | 38.0 | 16.7 |
| | ✓ | | 56.1 | 35.3 |
| | | ✓ | 56.4 | 37.9 |
| | ✓ | ✓ | 61.9 | 43.0 |
| ✓ | ✓ | ✓ | 62.1 | 43.2 |
| ✓ | ✓✓ | ✓ | **63.3** | **44.2** |

(b) Ablation on **Feature Extraction**.

| Feature Extraction | Pascal-$5^i$ | COCO-$20^i$ |
|---|---|---|
| | mean IoU | mean IoU |
| After MSDeformAttn | 60.5 | 42.8 |
| Before MSDeformAttn | **63.3** | **44.2** |

(c) Ablation on **Training Strategy**.

| Training Strategy | Pascal-$5^i$ | COCO-$20^i$ |
|---|---|---|
| | mean IoU | mean IoU |
| End-to-end | 60.6 | 40.3 |
| **2-stage-training** | **63.3** | **44.2** |

Table 4: **Ablations on Potential Objects Segmenter (Stage-1)**.

(a) Ablation on **Mask classification**.

| Mask classification | Pascal-$5^i$ | COCO-$20^i$ |
|---|---|---|
| | mean IoU | mean IoU |
| ✓ | 78.9 | 68.4 |
| × | **82.5** | **68.9** |

(b) Ablation on **Number of Segmenters**.

| Num Segmenters | Pascal-$5^i$ | |
|---|---|---|
| | Oracle | Final |
| 10 | 70.4 | 57.9 |
| 50 | 78.2 | 61.0 |
| **100** | 82.5 | **63.3** |

**Results on COCO-$20^i$.** In Tab. 1, our MM-Former performs remarkably well in COCO for both 1-shot and 5-shot setting. Specifically, we achieve $3.9\%$ improvement on 1-shot compared with CyCTR [34] and outperform HSNet [19] by $1.5\%$ mIoU .

**Results on Pascal-$5^i$.** Due to the limited number of training samples in the Pascal dataset, the POS may easily overfit during the first training stage. Therefore, following recent works [24, 1, 19], we include the transferring results that transfer the COCO-trained model to be tested on Pascal. Note that when training on the COCO dataset, the testing classes shared with the Pascal dataset are removed, so as to avoid the category information leakage.

According to Tab. 2, although our MM-Former is slightly inferior to some competitive results when training on Pascal dataset, we find MM-Former exhibits remarkable transferability when training on COCO and testing on Pascal. Specifically, the previous state-of-the-art method HSNet shows powerful results on Pascal→Pascal but degrades when transferring from COCO to Pascal. Instead, our MM-Former further enhances the performance of 1-shot and 5-shot by $4.4\%$ and $5.5\%$, outperforming HSNet by $6.1\%$ and $1.7\%$, respectively.

**Oracle Analysis.** We also explore the room for further improvement of this new few-shot segmentation paradigm by using query ground truth (GT) during inference. The results refer to the last rows in Tab. 1, 2. In detail, after the POS generates N mask proposals, we use the GT mask to select one proposal mask with the highest IoU, and regard this result as the segmentation result. Note that this natural selection is not the optimal solution because there may be other masks complementary to the selected one, but it is still a good oracle to show the potential of the new learning paradigm. According to the results, there is still a large gap between current performance and the oracle ($\approx 20\%$ mIoU), which suggests that our model has enormous potential for improvement whereas we have achieved state-of-the-art performance.

## 3.4 Ablation Studies

We conduct ablation studies on various choices of designs of our MM-Former to show their contribution to the final results. Component-wise ablations, including the MM Module and the POS, are shown in Sec. 3.4.1. The experiments are performed with the 1-shot setting on Pascal-$5^i$ and COCO-$20^i$. We further experimentally demonstrate the benefits of the two-stage training strategy in Sec. 3.4.2.

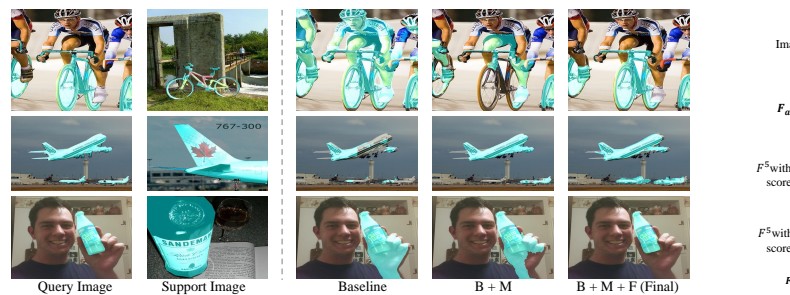

| Query Image | Support Image | | Baseline | B + M | B + M + F (Final) |

Figure 4: **Qualitative results on Pascal-$5^i$**. B, M, F mean Baseline network, learnable matching block and feature alignment block.

Figure 5: Visual results in Self Alignment block.

#### 3.4.1 Component-wise Ablations

To understand the effect of each component in the MM module, we start the ablations with a heuristic matching baseline and progressively add each building block.

**Baseline.** The result of the heuristic matching baseline is shown in the first row of Tab. 3a, which directly selects the mask corresponding to the highest cosine similarity with the support prototype. Note that when not using the learnable matching block, the results in Tab. 3a are all obtained in the same way as the heuristic matching baseline. We observe that the heuristic matching strategy does not provide strong performance, which is caused by the feature misalignment problem and fails to fuse multiple mask proposals.

**Self-Alignment Block**. With the self-alignment block, the performance is improved by $4.3\%$ on Pascal and $1.2\%$ on COCO, as shown by the $2^{nd}$ result in Tab. 3a, demonstrating that channel-wise attention does help normalize the features for comparison. However, the performance is still inferior, encouraging us to further align the support and query features with the cross-alignment block.

**Cross-Alignment Block**. In the third result of Tab. 3a, we experiment with a non-parametric variant of the cross-alignment block that removes all learnable parameters in the cross-alignment block. A significant performance drop is observed. This is not surprising because the attention in the cross-alignment block cannot attend to proper areas due to the feature misalignment. When learning the cross-alignment block, indicated by the $4^{th}$ result of Tab. 3a, the performance is remarkably improved by $14\%$ on Pascal and $12.9\%$ on COCO, manifesting the necessity of learning the feature alignment for further matching.

**Learnable Matching Block**. Surprisingly, simply using our learnable matching block can already achieve decent performance (the $5^{th}$ result in Tab. 3a) compared with the baseline, thanks to its capability to adaptively match and merge multiple mask proposals.

**Mask Matching Module**. By applying all components, our MM-Former pushes the state-of-the-art performance on COCO to $43.2\%$ (the $7^{th}$ result in Tab. 3a). In addition, to encourage the alignment of query and support in the feature space, we add the auxiliary loss $L_{co}$ to the output of the cross-alignment block, which additionally enhances the performance by more than $1\%$.

**Potential Objects Segmenter** Although we follow Mask2former to build our POS, several differences are made and we evaluate the choice of design as follows. *Mask Classification*. In Mask2Former [4], a linear classifier is trained with cross-entropy loss for categorizing each mask proposal, while in our MM-Former for few-shot segmentation, we remove it to avoid learning class-specific representations in the first training stage. The result in Tab. 4a shows that the classifier harms the performance due to that the linear classifier would make the network fit the "seen" classes in the training set. Since this change only affects the first stage, we use the oracle results to demonstrate the effect. *Numbers of Proposals*. In Tab. 4b, we try to vary the numbers of the mask proposals $N$. Increasing the number of $N$ will significantly improve the oracle result and our result. Thus we chose 100 as the default value in all other experiments. It is worth noting that, when varying the number from 10 to 100, our result is improved by $5.4\%$, but the oracle result is improved by $12.1\%$, indicating the large room for improvement with our new mask matching paradigm.

**Effect of Different Feature Extraction**. Previous few-shot segmentation works [34, 24, 30] typically integrate matching operations with segmentation-specific techniques [39, 24, 3]. Following Mask2Former, our POS also includes a multi-scale deformable attention (MSDeformAttn) [39]. In

Table 5: Analysis of the model efficiency, training time for MM-Former is shown in terms of $1^{st}$ / $2^{nd}$ stage.

| | 1-shot | | | | 5-shot | | | |
|---|---|---|---|---|---|---|---|---|
| | mIoU | training time | GPUs | memory | mIoU | training time | GPUs | memory |
| HSNet | 39.2 | 168h | 4 | 27.5G | 46.9 | 168h | 4 | 27.5G |
| CyCTR | 40.3 | 32.6h | 4 | 115.7G | 45.4 | 56.6h | 4 | 136.6G |
| MM-Former | 44.2 | 7.1h / 4.0h | 1 | 10.7G / 6.0G | 48.4 | 7.1h / 8.6h | 1 | 10.7G / 11.3G |

Tab. 3b, we investigate using the features from the MSDeformAttn instead of the backbone feature for the MM module. Interestingly, although the feature after the context modeling is essential for segmentation, it is not suitable for the matching problem and impairs the matching performance.

### 3.4.2 Analysis of Training Strategy

**Effect of Two-stage Training**. One may wonder what if we couple the training of POS and MM modules. Tab. 3c experiments on this point, the joint optimization is inferior to the two-stage training strategy, since POS and MM have different convergence rates.

**Efficiency of Two-stage Training**.We analyze the efficiency of our method and provide comparisons of training time, and training memory consumption (Tab. 5). All models are based on the ResNet-50 backbone and tested on the COCO benchmark. All models are tested with RTX A6000 GPU(s) for a fair comparison. Training times for CyCTR and HSNet are reported according to the official implementation. We report the training time for the first and second stages separately. It is worth noting that for the same test split, our method can share the same stage-1 model across 1-shot and 5-shot. The training time of stage-1 for 5-shot can be ignored if 1-shot models already exist.

### 3.5 Analysis of model transferability

Our MM-Former shows a better transfer performance when trained on COCO but a relatively lower performance when trained on Pascal. We make an in-depth study of this phenomenon.

**Effect of the number of training samples**: We use all training samples belonging to 15 Pascal training classes from COCO to train MM-Former. In this case, training samples are 9 times larger than the number in Pascal but the categories are the same, dubbed COCO-15 in Tab. 6. When the number of classes is limited, more training data would worsen the matching

Table 6: **Transfer** study (testing on Pascal).

| Training Set | 1-shot | 5-shot | Oracle |
|---|---|---|---|
| Pascal | 63.3 | 64.9 | 82.5 |
| COCO-15 | 60.7 (-2.6) | 64.8 (-0.1) | 86.3 |
| COCO-75-sub | 66.8 (+3.5) | 68.9 (+4.0) | 85.3 |
| COCO | 67.7 (+4.4) | 70.4 (+5.5) | 85.8 |

performance (60.7% vs. 63.3% for 1-shot and 64.8% vs. 64.9% for 5-shot), though a better POS could be obtained, as indicated by the oracle result (86.3% vs. 82.5%).

**Effect of the number of training classes**: We randomly sample an equal number of training images ( 6000 images averaged across 4 splits) as in Pascal training set from 75 classes (excluding test classes) in COCO to train our MM-Former, dubbed COCO-75-sub in Tab. 6. When training with the same amount of data, more classes lead to better matching performance (66.8% vs. 63.3% for 1-shot and 68.9% vs. 64.9% for 5-shot).

In a word, the **number of classes** determines the quality of the matching module. This finding is reasonable and inline with the motivation of few-shot segmentation and meta-learning: learning to learn by observing a wide range of tasks and fast adapting to new tasks. When the number of classes is limited, the variety of tasks and meta-knowledge are restricted, therefore influencing the learning of the matching module.

### 3.6 Qualitative Analysis

**Visual examples**: We show some visual examples in Fig. 4. Without loss of generality, some support images may only contain part of the object (*e.g.*, the $2^{nd}$ row). Directly selecting the mask with the highest cosine similarity can not obtain the anticipated result. Using a learnable mask matching block to fuse multiple masks can solve the problem to a large extent, the proposed feature alignment block can further improve our model by alleviating the misalignment problem, *e.g.* the results in the last row.

**Robustness analysis**: We also provide robustness analysis in Fig. 6, which uses an anomaly support sample for segmenting the query image. Compared with the previous state-of-the-art method, HSNet, which tends to segment the salient object in the image, our model is more robust to the anomaly inputs.

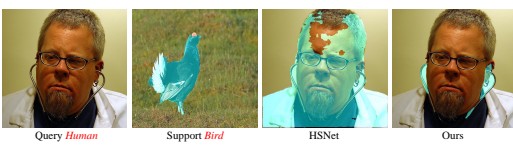

Figure 6: **Robustness Analysis.** We use different classes of support and query images to verify the robustness of the network.

**Explanation of SA**: SA is proposed to align the features at the channel dimension so that outliers at the channel dimension could be smoothed and features would be more robust by aligning with the attended global (specifically, channel-wise weighed average) features. Fig. 5 proves this point. It can be seen that $F_{avg}$ (row $2^{th}$) has response to general foreground regions. Important channels (row $3^{th}$) are emphasized, and outliers are suppressed (row $4^{th}$).

## 4    Related Work

**Few-Shot Segmentation** [21] is established to perform segmentation with very few labeled images. Many recent approaches formulate few-shot segmentation from the view of metric learning [23, 7, 27]. PrototypicalNet [22] is the first to perform metric learning on few-shot segmentation. PFENet [24] further designs an feature pyramid module to extract features from multi-levels. Many recent methods point out that only a single support prototype is insufficient to represent a given category. To address this problem, [32] attempt to obtain multiple prototypes via EM algorithm. [15] utilized super-pixel segmentation technique to generate multiple prototypes. Another way to solve the above problem is to apply pixel-level attention mechanism. [32, 26] attempt to use graph attention networks to utilize all foreground support pixel features. HSNet [19] propose to learn dense matching through 4D Convolution. CyCTR [38] points out that not all foreground pixels are conducive to segmentation and adopt cycle-consistency technology to filter out proper pixels to guide segmentation.

**Transformers** originally proposed for NLP [25] are being rapidly adapted in computer vision task [8, 2, 29, 5, 4]. The major benefit of transformers is the ability to capture global information using self-attention module. DETR [2] is the first work applying Transformers on object detection task. Mask2Former [4] using Transformers to unify semantic segmentation and instance segmentation. Motivated by the design of MaskFormer, we apply transformers to segment all potential objects in one image, align support features and query features in pixel-level within our MM-Former.

## 5    Conclusion

In the paper, we present Mask Matching Transformer (MM-Former), a new perspective to tackle the challenging few-shot segmentation task. Different from the previous practice, MM-Former is a two-stage framework, which adopts a Potential Objects Segmentor and Mask Matching Module to first produce high-quality mask proposals and then blend them into the final segmentation result. Extensive experiments on COCO-$20^i$ and Pascal-$5^i$ well demonstrate the effectiveness and the generalization advantage of the proposed MM-Former. We hope our MM-Former can serve as a solid baseline and help advance the future research of few-shot segmentation.

**Limitations and societal impact.** Our MM-Former introduces the paradigm of *decompose first and then blend* to the research of few-shot segmentation, which is a totally new perspective and may inspire future researchers to develop more advanced versions. However, there is still a large gap between the current results and the oracle ($\approx 20\%$ mIoU). How to further narrow this gap is our future research focus.

**Acknowledgment.** This work was supported in part by the National Key R & D Program of China (No.2021ZD0112100), the National NSF of China (No.U1936212, No.62120106009), the Fundamental Research Funds for the Central Universities (No. K22RC00010). Yao Zhao and Yunchao Wei are the corresponding authors.

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
