# OpenReview forum: "Mask Matching Transformer for Few-Shot Segmentation"
_NeurIPS.cc/2022/Conference — NeurIPS 2022 Accept_

### Official Review · Reviewer_Ytwe · 2022-06-28

**Rating:** 6
**Confidence:** 4
**Soundness:** 3 good
**Presentation:** 3 good
**Contribution:** 3 good

**Summary:**

The authors propose an alternate way of performing FSS by a 2-stage training procedure, where a network is first trained to generate a set of potential masks for the query image, followed by a second stage where another network is trained to generate features for both query and support, which are then pooled using the support masks, and the candidate query masks. From this, they find the best matching candidate masks through correlation followed by an MLP. The authors claim that the decoupling of segmentation and matching is beneficial for learning, and to back it up they show strong experimental results.

**Questions:**

## Questions regarding configs and code
Following the authors instructions in the code readme:

> We conduct a 2-stage-training strategy.
The Potential Objects Segmenter (POS) is trained in stage 1. The config files for PASCAL and COCO are provided at
"configs/pascal/step1.yaml" and "configs/coco/step1.yaml"
The Mask Matching Module (MM) is trained in stage 2. The config files are provided at
"configs/{*datasets*}/step2_{*split*}.yaml"
Execute this command at the root directory:
    python train.py --config-file {*model_config*}
Note: We recorded the random_seed during training, using this random seed can fully reproduce the results in the paper.

Notably the authors only provide a single config for the POS network. In the configs it seems like there should be one POS model per fold, so this might have been a mistake.

If the stage 1 training involves a POS model which sees classes belonging to the test set of other folds, there is leakage.

## Question regarding loss for POS during stage 1
In MaskFormer, the loss is divided into classification and segmentation of all K classes. Typically in episodic FSS training the given label is only of the single mask of the target class. Is this also the case in your method, or is your segmentation loss based on all classes that appear in the image? This was not obvious to me from the paper, and I could also not really understand it from reading the code, as the code is quite complex.

## Question regarding generalization of POS
If there is no class leakage between base and novel classes during training of POS, how come the N proposals don't overfit to the base classes? In the paper it is claimed that the frozen backbone keeps the POS from overfitting, but one could easily see the proposals closely matching the base classes in feature space since the queries are learnable. Is there any mechanism to prevent this from happening?

## Question regarding motivation for 2-stage training
In the paper you claim that training end-to-end:
> When training in an end-to-end fashion, POS would be influenced by the training of
the MM module, degrading the quality of mask proposals

I'm not sure I follow the reasoning behind this. Could you provide a more thorough explanation why one would expect the masks to become worse in this case? From what I gather you seem to be saying that focusing only on the segmentation loss is beneficial, since propagating other losses could lead to worse segmentations. However, why not make a seperate mask prediction for the matching and segmentation tasks, but still corresponding to the same proposal?

## Anonymization
The provided code is not sufficiently anonymized, I found references to author home folders in the code. (I did not google the names to respect reviewing anonymity). For future submissions I recommend the authors to more thoroughly anonymize the code.



**Limitations:**

The authors' usage of an oracle model to see the performance gap was an interesting and useful way to gauge the current limitations.

**Strengths And Weaknesses:**

## Major Strengths
* Novel approach
* Simple architecture (although 2-stage training makes it more complicated)

## Minor Strengths
* Good results
* Good ablations
* Code provided

## Major Weaknesses
* Concerns for test-time class leakage into the first stage of training.

## Minor Weaknesses
* Somewhat underwhelming results on PASCAL
* Even without class leakage, I'm somewhat unconvinced of the reasoning behind the improvement from 2-stage training.
* The paper is somewhat difficult to follow
* The code is hard to read and seems to include multiple mistakes

## Conclusion
Without further clarification by the authors I can not recommend the paper in its current form for acceptance, hence I've selected a rating of 4. However, I am open to increasing the rating to a 6 or 7 given that the authors address my concerns and questions.

### EDIT
After the response of the authors I am increasing my rating from a 4->6, and confidence from 3->4.

---

> ### Author Response · Authors · 2022-08-02
> **Response to Reviewer Ytwe**
>
> Thank you so much for acknowledging the strength of our method. We have carefully considered your constructive and insightful comments and here are the answers to your concerns.
>
> **Q1. Concerns for information leakage in the first stage.**
> We want to clarify that for stage-1 POS training, we train different POS models for different splits, and there is **NO** test-time class information leakage in the first stage. We only provide one config file "step1.yaml" as an example, but we adjust the split-related parameter in the config when training POS for each split. This implementation detail is also stated in **Line 207** of the main text.
>
>
> **Q2. Loss function for POS**
> Mask2former uses binary mask loss for segmentation and multi-way classification loss for mask classification. In our POS training, we remove the classification loss (Line 129) but only use the binary mask loss. Besides, different from the episodic training that randomly selects one target class for training, we train our POS based on all base classes (excluding test-time novel classes). File {*mask2former/data/datasetmappers/fewshotmapperori.py*} shows how to generate binary masks of base classes from semantic labels.
>
>
> **Q3. Generalization of POS**
> As stated in Q2, our POS is only trained with binary mask loss so that it is a class-agnostic segmentation model that can also produce segmentation masks for objects of unseen classes. This idea is similar to the Region Proposal Network (RPN) in object detection, which shows great potential to provide proposals for unseen classes and can be used for Zero-shot/Open-vocabulary object detection [A].
>
>
> **Q4. Motivation for 2-stage training**
> From the optimization side, as you conjectured in the review, the loss of MM has a negative effect on training POS. On the one hand, POS and MM have different convergence rates. For instance, POS requires more than **12** epochs to converge, but MM only needs just **2** epochs. On the other hand, MM is optimized with episodic training and supervised by one of the target classes, while POS requires maintaining the general segmentation ability. Therefore, gradients from MM would interfere with the learning of POS. We empirically verify this point in Tab.3$(c)$ where 2-stage training improves the performance by 2.7\% and 3.9\% on Pascal and COCO, respectively, compared with the end-to-end training.
>
> Besides, from the training efficiency side, most FSS methods require to re-train the whole model for different “shots”, *e.g.* 1-shot and 5-shot in FSS. Our 2-stage training allows us to share the first stage for different “shots”, which largely improves the training efficiency. Please refer to **General Response-Q1** for the efficiency comparison and discussion. Our approach only requires additional 4.0h and 8.6h wall-clock time for training 1-shot and 5-shot models on the COCO dataset with only one GPU.
>
>
> [A] Open-vocabulary Object Detection via Vision and Language Knowledge Distillation, Xiuye Gu et al. ICLR'21.

---

> > ### Comment · Reviewer_Ytwe · 2022-08-06
> > **Information leakage, loss functions , generalization, motivation**
> >
> > ## Regarding Information Leakage
> > I will take the authors explanation by their word. It is unfortunate that the submitted code is somewhat unclear and difficult to read. I recommend the authors improve code readability.
> >
> > ## Loss Functions
> > I thank the authors for clarifying these points.
> >
> > ## Generalization
> > While I understand that the network is trained with binary classification, that does not change the underlying fact that the mask proposals are in feature space and can therefore adapt to overfit the base classes. I believe that the relatively larger improvements on COCO compared to PASCAL might be attributed to this.
> >
> > ## 2-stage training motivation
> > I acknowledge the arguments made by the authors regarding different convergence rates. While I'm not entirely convinced that 2-stage training is the best solution, I think the authors have made good arguments for their approach.
> >
> > ## Concerns of other reviewers
> > Other reviewers expressed concerns that the proposed method is relatively weaker in the low training data regime and therefore not good for few-shot segmentation. I would argue the opposite. In my opinion, few-shot methods are meant to be trained on larger datasets to then adapt to small datasets at inference time. Hence I believe COCO to be a more suitable training dataset, and more emphasis ought to be put on those results.
> >
> > ## Conclusion
> > I believe the authors rebuttal sufficiently address my concerns, and am therefore increasing my rating from 4 to 6.

---

> > > ### Author Response · Authors · 2022-08-06
> > > **Thank you for your recognition and suggestions**
> > >
> > > Thanks for the valuable discussion and suggestions. We will make sure to organize the code to improve the readability before open-source, as well as continue exploring better mask-matching solutions beyond the 2-stage training as future work.

---

### Official Review · Reviewer_Ecav · 2022-07-09

**Rating:** 4
**Confidence:** 5
**Soundness:** 2 fair
**Presentation:** 3 good
**Contribution:** 2 fair

**Summary:**

This paper proposes a few-shot semantic segmentation (FSSS) network based on Mask2Former [1]. By formulating the semantic segmentation as a mask classification problem like [1], the authors introduced a new matching mechanism in mask-level instead of pixel-level matching in previous FSSS methods. Additionally, they propose a feature-alignment block based on the attention mechanism to align both support and query features individually and cross-align between them. They conducted experiments on two FSSS datasets (COCO-20i and PASCAL-5i) and achieve the SOTA results.

**Questions:**

+ Can the author add the running time, training time, and the memory consumed by the model? As the Mask Matching Module is implemented by standard transformer decoder layers, does it requires large memory and time?
+ What is the detailed implementation of the Mask Matching Module?
In Fig. 2, the final mask has the same resolution as the F2 feature map, how to upsample the mask to the original resolution of the input image?
+ Explain the self-alignment module and intuition of this. Can we replace it with the self-attention module (compare the performance and runtime)
+ Explain the use of the MLP in Eq.(4): describe the input, and output range, and how it can correct the cosine similarity vector of N potential masks. Visualize the M and M^ before and after. It helps the reader what is happening inside the block.
Why do not use dice loss in L_m (L173)
+ L180: “Meanwhile, only constrain on Sˆ pos may lead all outputs of Cross Alignment block to tend to be same”: why? explain in more detail?
+ L182: how to find the lowest IoU (ex: 2 potential masks with no overlap with GT)
+ Can the authors add more visualization of the failure cases?
+ In Eq. (4), redundant closing parenthesis ')'.

I will increase my score if the authors address all of my concerns.

**Strengths And Weaknesses:**

**Strengths**
+ The structure of this paper is well-organized and easy to follow the ideas of the authors.
+ The proposed method achieves significant improvements on the two datasets compared with other recent methods.

**Weaknesses**
+ The only novel contribution of this paper is the Feature Alignment Block (FAB) in Mask Matching Module and it significantly improves the results, the other blocks are incremental. The authors simply adapt it to Mask2Former architecture. It would be better to investigate more about the FAB in different methods (CyCTR, HSNet) to prove its effectiveness is generalizable enough.
+ The “few-to-few” matching (L53) is not new. Previous approaches ([2], [3]) have applied it to the few-shot instance segmentation task.
The proposed method (the mask2former architecture) tends to overfit small datasets (L225), then it is not suitable for the few-shot setting.
+ Related work is not well presented. They do not provide any analysis or comparison of previous work.

---

> ### Author Response · Authors · 2022-08-02
> **Response to Reviewer Ecav ( 2/2 )**
>
> **Q4. Questions for Self Alignment(SA) Module**
> * **Intuition of SA:** Note that our Mask Matching Module only focuses on the matching problem rather than the segmentation problem. For the segmentation problem, we intuitively use spatial-wise self-attention because it aligns each pixel with the attended global (specifically, spatial-wise weighted average) representation, and the pixel-level outliers would be smoothed. On the contrary, for the mask matching problem, which matches masks by comparing corresponding feature vectors, we need to align the features at the channel dimension so that outliers at the channel dimension could be smoothed and features would be more robust by aligning with the attended global (specifically, channel-wise weighed average) features.
>
> * **Explanation of SA**: The input feature map $F \in \mathbb{R}^{c \times hw}$ is first averaged at the channel dimension to obtain  $F_{avg} \in \mathbb{R}^{1 \times hw}$. Then, $F_{avg}$ is regarded as an anchor to obtain the attention weight $A \in \mathbb{R}^{c \times 1}$ by matrix multiplication: $A = F F^T_{avg}$, which represents the weights of different channels. Finally, $A$ is used to activate the feature by position-wise-multiplication (*i.e.*, expand at the spatial dimension and perform point-wise multiplication with the feature) to obtain $F_{A} \in \mathbb{R}^{c \times hw}$. In this way, the input feature is adjusted across the channel dimension, and the outliers are expected to be smoothed. We have updated Fig. 3 with the mentioned symbols in the revision. Besides, to help understand, we provide some visualizations in Fig. **5** in revision that show the feature maps within this procedure. Specifically, we visualize 1). $F_{avg}$, 2) $F^{argmax(A)}$ and 3）$F^{argmin(A)}$ in Fig. 5. It can be seen that $F_{avg}$ has response to general foreground regions but $F^{argmin(A)}$ exhibits unstructured patterns.
>
> * **Ablation study of SA**: The difference between SA and self-attention has been analyzed in Point1 above. Following your suggestion, we replace SA with a Non-Local [B] self-attention block, measuring performance and computational cost in terms of mIoU and FLOPs as follows:
>
> |          | mIoU     | FLOPs |
> | -------- | -------- | -------- |
> | Self-Alignment module | 44.2     | 14.2K     |
> | Self-Attention module | 42.7     | 5.7G     |
>
>
> **Q5. Generalization of FAB**
> We apply our Feature Alignment Block (FAB) to CyCTR and HSNet to verify the generalization ability of FAB.
> * CyCTR+FAB: We directly insert FAB to CyCTR before the cycle-consistent transformer. With our FAB, the well developed CyCTR can still be improved by 0.3\% mIoU (from 64.0\% to 64.3\%). Besides, following Tab.5 in the CyCTR paper, where they ablate the number of cycle-consistent transformer encoders with 128 hidden dimensions. According to their results, using one more cyc-encoder only provides 0.2\% improvement (from 63.5\% to 63.7\%). With our FAB, CyCTR-128 can be improved by 0.6\% mIoU (from 63.5\% to 64.1\%).
>
> * HSNet+FAB: HSNet takes 13 middle-level output features from the backbone to the decoder. Performing Cross Alignment (CA) block at all 13 levels is impossible. Thus we only apply SA to HSNet. Due to the time and resource limitation, we only train HSNet+SA for 200 epochs, rather than the 2000 epochs suggested by the official implementation of HSNet. SA brings 0.2% mIoU improvement to HSNet. A longer training may lead to further improvement.
>
>
> |      |    CyCTR          | CyCTR-128 | HSNet |
> | ---- | ------------------- | ----- | ---   |
> | Original | 64.0           | 63.5  | 64.0  |
> | With FBA | 64.3            | 64.1  | 64.2  |
>
>
> **Q6. Analysis or comparison of previous work and visualization of failure cases**
> Due to space limitations, we analyzed the previous works in the introduction, and compress the related work. We will add analysis and comparison of previous works in our final version and add more visualizations of failure cases.
>
> [A] Meta r-cnn: Towards general solver for instance-level low-shot learning. Yan X, et al. ICCV 2019.
> [B] Non-Local Neural Networks, Wang X et al. CVPR2018.

---

> ### Author Response · Authors · 2022-08-02
> **Response to Reviewer Ecav ( 1/2 )**
>
> Thank you so much for acknowledging the strength of our method. We have carefully considered your constructive and insightful comments and here are the answers to your concerns.
>
> **Q1. The only novel contribution of this paper is the Feature Alignment Block.  The “few-to-few” matching (L53) is not new.**
> Thanks for acknowledging the novelty of our FAB. However, we would like to re-emphasize other contributions and novelty that are also new to few-shot segmentation.
> First, for few-shot segmentation, where previous works entangled the matching and segmentation tasks and were shown to be hard to be optimized, we are the first work that proposes the idea to decouple the segmentation and matching tasks.
> Besides, we find that a two-stage training strategy is beneficial for simplifying the optimization, and it is shown to be efficient for training (the efficiency analyses are provided in **General Response-Q1**.)
> In addition, we find MaskFormer/Mask2former is suitable for serving as the Potential Objects Segmenter to generate mask proposals for **unseen** classes, while it is possible to use other class-agnostic segmenter to generate mask proposals and would be an interesting future work for few-shot segmentation.
>
> Thanks for mentioning some few-shot instance segmentation works in the review, but we do not see the reference. We infer that the related work you mentioned would be Meta R-CNN [A]. Although Meta R-CNN uses meta learning to learn the predictor head, it cannot be applied to few-shot segmentation since it only focuses on instance-level prediction.
>
>
> **Q2. Analysis of model efficiency**
> Thanks for your suggestion. We provide a thorough analysis of the model efficiency in **General Response-Q1**.
> Note that benefiting from our decoupled design, the Mask Matching Module is easy to train and does not require large memory and time. Instead, it requires much less training time since all other parts of the network (backbone and segmentation decoder) are fixed, and the MM module is lightweight.
>
> **Q3. Technical details clarification.**
> * **Clarification of MM module and Fig2**
> In Cross Alignment (CA) module, we reduce the number of feature channels to 256. Setting the transformer layers to 2, hidden dimension of FFN to 512. The resolutions of matrix $k$, $v$ in Eq.(3) are downsampled to 1/32 of the original image size to further reduce computation. The Learnable Matching (LM) module in MM contains one MLP layer with a hidden layer of 500 nodes.
> The Final mask in Fig2  has the same resolution as the F2 feature map, we apply bilinear interpolation to upsample the final mask to the original resolution.
>
> * **Clarification of the MLP in Eq.(4) and visualization of $M$ and $\hat{M}$**
> In Eq.(4), the input of MLP, $S \in \mathbb{R}^{1 \times N}$, denotes N values of cosine-similarity between support prototype and N query prototypes. We can use $S_{out}$ to represent the output of MLP and $S_{out} \in \mathbb{R}^{1 \times N}$. You can regard $S_{out}$ as N weights of N mask proposals. A weighted average is applied to generate the final mask.
> We have explained the meaning of $M$ and $\hat{M}$ in Line 127 and Line 165 in the paper, $M$ means the N mask proposals, and $\hat{M}$ means the final mask. Some visualizations of $M$ and $\hat{M}$ are provided in Fig.1 and Fig.2.
> For the loss function of MM ($l_{m}$), both Dice Loss and BCE Loss have been tried and got similar results.
>
> * **Explanation of Line 180: Why only constraining $\hat{S}$ pos is not enough?**
> If we only optimize $\hat{S}$ pos towards $y_{pos}$, there may be a trivial solution that all outputs collapse to the same with the loss equal to zero (minimum). Therefore, we need to provide negative samples to prevent this trivial solution.
>
> * **Explanation of the lowest IoU in Line 182**
> In L182, We denote the $i^{th}$ mask proposal as $m_i$ and groundtruth as $gt$. IoU value is calculated by the following formula:
> \begin{equation}
> \mathrm{IoU}(m_{i}, gt) = \frac{\sum m_{i}*gt}{\sum {m_i}+ \sum {gt} - \sum m_{i}*gt },
> \end{equation}
> where $gt$ is a binary map, while $m_{i}$ takes value between 0 and 1. $\mathrm{IoU}(m_{i}, gt)$ will not be 0 because $m_i$ is continuous $gt$ is discrete. Thus one lowest IoU always exists.

---

> ### Comment · Reviewer_Ecav · 2022-08-08
> **My reponses**
>
> What do you mean by: "instead, it requires much less training time since all other parts of the network (backbone and segmentation decoder) are fixed"
>
> I can understand the backbone pretrained on the Imagenet dataset, but I don't understand the segmentation decoder here. It is pretrained on what dataset?

---

> > ### Author Response · Authors · 2022-08-08
> > **Clarification of the "segmentation decoder".**
> >
> > Thanks for the further comments.
> > The segmentation decoder here refers to the Potential Objects Segmenter (POS) module. We utilize a two-stage training strategy that trains the POS in the first stage. Our POS is trained on the same datasets, *i.e.,* Pascal and COCO, for each split, with test classes removed. This implementation detail is also stated in Line 206-207 of the main text and also explained in our response to Reviewer-Ytwe-Q2.

---

### Official Review · Reviewer_gY9J · 2022-07-11

**Rating:** 3
**Confidence:** 5
**Soundness:** 2 fair
**Presentation:** 2 fair
**Contribution:** 2 fair

**Summary:**

This paper proposes a new two-stage framework that decouples the matching and segmentation modules for few-shot segmentation. Extensive experiments and ablation studies on COCO and Pascal datasets also verify the algorithm's effectiveness.

**Questions:**


-  In #40,  " Concretely, pixel-level relationships are modeled between the support and query feature maps either by attention mechanism or 4D convolutions." Using "either...or..." is too arbitrary. There are also other methods to deal with the relationship between the query and support sets, like HSNet and PGNet.



- Poor performance on the Pascal dataset even compared with the methods in the community proposed years ago. In #85, the author attributed the reason to the limited scale of the dataset, which does not convince the reviewer.



- Adding more visual illustrations will make it clearer while introducing the network structure in subsection 2.2.


- In the Introduction and Abstract sections, the author emphasizes the advantages of the algorithm in complexity. Relevant comparative experiments and explanations need to be supplemented to support the claims.


-  In #115, " We use the outputs from the last three layers in the following modules", "the last three layers" is ambiguous. Please further elaborate on the details.


- The novelty is marginal since the core idea is borrowed from MaskFormer.


- The impacts of different components brought to the model efficiency should be discussed, including the overall final model efficiency in terms of fps and model size.


- There are quite a lot of grammatical errors and ambiguous expressions. The reviewer did not try to point them all out together. Please check your manuscript carefully and correct all mistakes before submission. Typos include but are not limited to:

a. In #22, "one of a fundamental tasks" -> "one of the fundamental tasks"

b. In #22, “ has achieved grand success” -> " has achieved a grand success"

c. In #29, " segment any objects" remove "any"

d. A large number of articles are missing or used incorrectly. For example, "We refer this kind of method as 'few-to-many' matching approach." -> "We refer this kind of method as a 'few-to-many' matching approach." I won't list all since there are too many.

e. In #37, "While acceptable results obtained" -> "While acceptable results were obtained". Lack of the predicate.

f. In #51, "Rather than matching" -> "Rather than being matched"

g. In #300, ", it is not suitable for the matching problem and impair the matching performance." -> "impairs"

**Limitations:**

The stated limitation, i.e., a large gap between the oracle case, indicates room for future improvements, the reviewer still believes that broader negative impacts and limitations should be discussed.

**Strengths And Weaknesses:**

1. The writing of this paper is redundant and ambiguous, and the logic is not clear.

2. There are lots of grammatical errors which need to be corrected.

3. Lack of innovation and low novelty.

4. Poor performance on the Pascal dataset even compared with the methods proposed years ago.

5. Proposed two-stage strategy introduces a new perspective focusing on mask-level segmentation.

---

> ### Author Response · Authors · 2022-08-02
> **Response to Reviewer gY9J**
>
> Thank you so much for acknowledging the strength of our method. We have carefully considered your constructive and insightful comments and here are the answers to your concerns.
>
> **Q1. The novelty is marginal since the core idea is borrowed from MaskFormer.**
> There may be some misunderstandings about the contributions of our method. We would like to re-emphasize the contributions and novelty as follows.
> First, for few-shot segmentation, where previous works entangled the matching and segmentation tasks and were shown to be hard to be optimized, we are the first work that proposes to decouple the segmentation and matching tasks.
> Besides, we find that a two-stage training strategy is beneficial for simplifying the optimization, and it is shown to be efficient for training (the efficiency analyses are provided in **General Response-Q1**).
>
> Note that we find MaskFormer/Mask2former is suitable for serving as the Potential Objects Segmenter to generate mask proposals for **unseen** classes, thus the core idea is different from that in MaskFormer/Mask2former. It is possible to use other class-agnostic segmenter to generate mask proposals and would be an interesting future work for few-shot segmentation.
>
>
> **Q2. Poor results on PASCAL-$5^{i}$**
> Although our model exhibit relatively lower performances when trained on the Pascal dataset, it shows a much better transfer performance from COCO to Pascal than other state-of-the-art methods. For instance, when compared with HSNet, our model obtains 67.7\% vs. 61.6\% for 1-shot and 70.4\% vs. 68.7\% in Tab. 2. We want to emphasize that since few-shot segmentation aims to obtain a class-agnostic model that can be used to segment objects of arbitrary unseen classes, transfer performance is also a practical illustration for measuring the effectiveness of the few-shot segmentation model.
> Besides, to make a clear understanding of the reason why our matching module performs relatively worse when being trained on Pascal, we provide a thorough analysis in **General Response-Q2** and reveal that the number of training samples is not the bottleneck for the matching module, but the number of the training classes is. We conjecture that other methods perform better when trained on Pascal (with few classes) because they entangle the segmentation and matching together and tend to segment the salient object in the image, as shown in Fig.6 in the revision, which deviates from the motivation of FSS. Please refer to **General Response-Q2** for a detailed discussion.
>
>
> **Q3. Model efficiency**
> The following table lists the inference time and parameters of different components. This table is an extension of Tab. $3(a)$ of our paper. The model size of the first line consists of the parameter sizes of the backbone network (23.2M) and the POS module (15.0M). We also provide a detailed efficiency analysis in **General Response-Q1** to support our claim that our approach has low training complexity (Line 89).
>
>
> | SA  | CA  | LM  | FPS  | model size |
> | --- | --- | --- | ---- | ---------- |
> |     |     |     |65.9 | 38.2M     |
> | ✓   |     |     |65.1  | + 0M      |
> |     | ✓   |     |52.0  |+ 2.5M |
> |     | ✓   | ✓   |51.2  |+ 2.6M|
> | ✓   | ✓   | ✓   |50.9 |+ 2.6M|
>
>
>
> **Q4. Ambiguous expression**
> For the ResNet backbone, we denote the outputs of the last residual blocks from conv2, conv3, conv4, and conv5 by {$F^{2}$, $F^{3}$, $F^{4}$, $F^{5}$}, where they have strides of {4, 8, 16, 32} with respect to the input image. By saying “the last three layers”, we refer to {$F^{3}$, $F^{4}$, $F^{5}$}. We will clarify this point in our final version.
>
>
> **Q5. Typos**
> Thank you for kindly pointing out the typos. We have carefully resolved the typos in the revision.

---

### Official Review · Reviewer_2Z5i · 2022-07-11

**Rating:** 5
**Confidence:** 4
**Soundness:** 3 good
**Presentation:** 3 good
**Contribution:** 2 fair

**Summary:**

This paper tackles the few-shot segmentation task through a proposed mask matching transformer (MM-Former). The MM-Former contains two parts. The first part decomposes query images into multiple segmentation proposals with a class-agnostic segmenter. The second part merges related segment proposals into final masks guided by support images. With the ResNet-50 backbone, the proposed method get competitive results on the popular COCO and PASCAL benchmarks.

**Questions:**

I am curious about whether the number of meaningful object proposals affects the final performance. There are also many other issues discussed in the weakness section.

**Ethics Review Area:**

["I don’t know"]

**Limitations:**

I didn't see any obvious negative societal limitations of this work.

**Strengths And Weaknesses:**

***Strengths***
1. This paper is well-organized and can be easily understood by readers. The technical details are introduced clearly.
2. The authors conducted extensive experiments on multiple benchmarks to investigate the effectiveness of different modules and designs in this paper.

***Weakness***
1. In the potential objects segmenter section, the authors just borrow the ideas from Mask2Former to generate object proposals. But in few-shot learning, the number of meaningful object parts is unknown both in the base training set and the novel test set. How does the network learn these object segmenter without any guidance? If the object segmenter is inaccurate, will it affect the subsequent query-related part merge?
2. The proposed MM-Former exploits transformers as decoders for few-shot semantic segmentation. As is known a transformer may need a lot of learnable parameters. It will be better to list the parameters of different modules in MM-Former to see how the number of learnable parameters of transformers affects the final performance.
3. In Table 2, we can find that the performance of MM-Former on PASCAL is not good as other state-of-the-art algorithms. I want to see a thorough analysis of such a phenomenon. Maybe we can know more about the characteristics of MM-Former.

---

> ### Author Response · Authors · 2022-08-02
> **Response to Reviewer 2Z5i**
>
> Thank you so much for acknowledging the strength of our method. We have carefully considered your constructive and insightful comments, and here are the answers to your concerns.
>
> **Q1. How does the network learn these object segmenters without any guidance?**
> To train the POS module, we remove the classification loss from Mask2Former but only use the binary mask loss for generating binary mask proposals. Therefore, it is a class-agnostic segmentation model that can produce segmentation masks for objects of unseen classes. This idea is similar to the Region Proposal Network (RPN) in object detection, which is capable of providing proposals for unseen classes and can be used for Zero-shot/Open-vocabulary object detection [A].
>
> **Q2. Concerns for Inaccurate/ Meaningful mask proposals.**
> * **Inaccurate mask proposals:** It is possible that POS may produce inaccurate mask proposals when the testing distribution is extremely different from the training distribution, for instance, training on natural images but testing on synthetic data. However, according to our oracle performance in Tab.1 and Tab.2 for Pascal and COCO datasets, the POS does show well generalization capability to unseen classes with high-quality proposals.
> To quantitatively study how inaccurate proposals influence the final performance, for each testing query image, we remove the mask that has the highest IoU with the ground truth mask before merging, and the results are shown in the table below. The quality of proposals has a large influence on the oracle result (-2.5\% mIoU) since the oracle chose the mask with the highest IoU as the result. Instead, our Mask Matching Module is more robust to the accuracy of the proposals, since we merge multiple masks as the final result.
>
> |     | Oracle mIoU| Final mIoU|
> | --- | ------ | ----- |
> | original | 82.5   | 63.3  |
> | remove best proposal | 80.0 (-2.5)  | 61.9 (-1.4) |
>
> * **Number of meaningful proposals:** It is hard to define the concept of "meaningful object parts". Moreover, the number of "meaningful object parts" is unknown as it is impossible to find a certain threshold for various query images. Therefore, as the table above shows, we study one extreme case that removes the "most meaningful" proposal for all query images to show the robustness of our matching module.
>
>
>
>
> **Q3. Number of learnable parameters/ How does the number of learnable parameters of transformers affect the final performance?**
> * First, the number of parameters of each part of our MM-Former is listed in the following table. The backbone is ResNet-50 and POS module follows Mask2Former except that we remove the classification head. Our Mask Matching Module is light-weighted and only introduces 2.6M parameters, which is only around 11\% of the size backbone network.
>
> | Backbone | POS Module  | MM Module  | Total|
> | -------- | ----- | --- | ---  |
> | 23.2M    | 15.0M | 2.6M| 40.8M|
>
> * Besides, we vary the learnable parameters in MM by adjusting the hidden dimension of FFN and the number of transformer blocks and show how they affect the final performance. We conduct experiments on COCO-$20^{i}$, where * denotes the default setting of our MM-Former, d is the hidden dimension of FFN and L is the number of transformer layers.
>
> |  (d, L)  | (256,1) |  (512,1) | (512,2)* | (768,3) | (1024,4) |
> | ---        | ---     | ---      | -------- | --------| -------- |
> | mIoU       |  43.4   |  43.2    |  44.2    |  42.8   |   42.8   |
> |Learnable Parameters|  1.4M   |  1.8M    |  2.6M    |  4.6M   |   7.3M   |
>
>
> **Q4. Analyzing the phenomenon of underperforming results on Pascal training but better transfer performance.**
> Thanks for your great suggestion. We provide a thorough analysis of this phenomenon in the **General Response-Q2**, in which we find that increasing the number of classes would benefit the learning of the matching module. Our decoupled design and the matching performance also encourage us to re-think the reasonability of current FSS benchmarks.
>
> [A] Open-vocabulary Object Detection via Vision and Language Knowledge Distillation, Xiuye Gu et al. ICLR'21.

---

### Author Response · Authors · 2022-08-02
**General Response**

**Q1. Analysis of model efficiency**
As most Reviewers have concerns about the efficiency of MM-Former, we analyze the efficiency of our method and provide comparisons of inference time (in terms of FPS), training time (estimated wall-clock time), and training memory consumption. We compare our method with HSNet and CyCTR using their public codes. All models are based on the ResNet-50 backbone and tested on the COCO-$20^{i}$ benchmark. Inference times for all models are tested with one single RTX A6000 GPU for a fair comparison. Training times for CyCTR and HSNet are reported according to the official implementation. Since we adopt the two-stage training strategy, we report the training time for the first and second stages separately. It is worth noting that for the same test split, our method can share the same stage-1 model across 1-shot and 5-shot. Therefore, the training time of the first stage for 5-shot can be ignored if 1-shot models already exist.

1-shot semgentation:
|   | mIoU | inference time (FPS)  | training time ($1^{st}$/$2^{nd}$ stage) | training GPUs |memory consumption ($1^{st}$/$2^{nd}$ stage)  |
| -- | -- | -- | --|--| -- |
| CyCTR | 40.3 | 29.1 |  32.6h | 4   |  115.7G |
| HSNet | 39.2 | 59.0 | 168h | 4   | 27.5 G|
| Ours  | 44.2 | 50.9 | (7.1h) / 4.0h|1 |  10.7G / 6.0G |

5-shot semgentation:
|  | mIoU | inference time (FPS)  |  training time ($1^{st}$/$2^{nd}$ stage)  | training GPUs |memory consumption ($1^{st}$/$2^{nd}$ stage) |
| -- | -- | -- |  --|--| --  |
| CyCTR | 45.4 | 11.3 | 56.6h | 4 | 136.6G |
| HSNet| 46.9 | 10.8 | 168 h | 4 | 27.5 G|
| Ours  | 48.4 |12.4 |  (7.1h) / 8.6 h | 1 | 10.7G / 11.3G  |


CyCTR and HSNet require to optimize segmentation and pixel-level matching together, causing a much longer training time compared with our decoupled design. Specifically, our POS only focuses on the class-agnostic segmentation problem and only needs about 12 epochs on COCO without episodic training, and Mask Matching (MM) module focuses on the matching problem and only requires 2 epochs of episodic training to converge.



**Q2. Results on Pascal**
Our MM-Former shows a better transfer performance when trained on COCO but relatively lower performance when trained on Pascal. Some reviewers have concerns about the effectiveness of our method and see this as a weakness. We want to emphasize that since few-shot segmentation aims to obtain a class-agnostic model, transfer performance is also a practical illustration for measuring the effectiveness of the few-shot segmentation model.

Besides, to provide a deep and thorough understanding of this phenomenon, as suggested by Reviewer 2Z5i, we consider the following two aspects that may influence the training performance:
* Effect of **the number of training samples**: We use all training samples belonging to 15 Pascal training classes from COCO to train our MM-Former. In this case, training samples are ~9 times larger than the number in Pascal but the categories are the same, dubbed *COCO-15*.
* Effect of **the number of training classes**: We randomly sample a equal number of training images (~6000 images averaged across 4 splits) as in Pascal training set from 75 classes (excluding test classes) in COCO to train our MM-Former, dubbed *COCO-75-sub*.


| Method | Training Set | 1-shot mIoU | 5-shot mIoU | Oracle mIoU |
| - | - | - | - | - |
|HSNet| Pascal  | 64.0 | 69.5  |-|
|HSNet| COCO-75  | 61.6 (-2.4) | 68.7 (-0.8) |-|
|MM-Former| Pascal | 63.3| 64.9 | 82.5  |
|MM-Former| COCO-15| 60.7 (-2.6) | 64.8 (-0.1) | **86.3** |
|MM-Former| COCO-75-sub| 66.8 (+3.5) | 68.9 (+4.0) | 85.3 |
|MM-Former| COCO-75 | 67.7 (+4.4) | 70.4 (+5.5) | 85.8 |

For comparison, results obtained with Pascal and 75 classes COCO data (dubbed COCO-75) from Table 2 of the manuscript are also included in this table. We also include the results of HSNet for comparison.

It is interesting to have the following observations:
* When training with the same amount of data, more classes lead to better matching performance (66.8\% vs. 63.3\% for 1-shot and 68.9\% vs. 64.9\% for 5-shot).
* When the number of classes is limited, more training data would **worsen** the matching performance (60.7\% vs. 63.3\% for 1-shot and 64.8\% vs. 64.9\% for 5-shot), though a better POS could be obtained, as indicated by the oracle result (86.3\% vs. 82.5\%).
* With sufficient training classes, more training data can benefit the matching performance (67.7\% vs. 66.8\% for 1-shot and 70.5\% vs 68.9\% for 5-shot).

In a word, we find that it is the **number of classes** determines the quality of the matching module. This finding is reasonable and inline with the motivation of few-shot segmentation and meta-learning: learning to learn by observing a wide range of tasks and fast adapting to new tasks. When the number of classes is limited, the variety of tasks and meta-knowledge are restricted, therefore influencing the learning of the matching module.

---

### Author Response · Authors · 2022-08-09
**Thank all reviewers, and welcome to discuss**

Dear Reviewers,

We thank all of you for taking your valuable time to provide insightful comments, which significantly strengthen our paper. We have carefully responded to your questions accordingly with the necessary additional experiments and analyses. We hope our responses have addressed all your concerns. Since the discussion period is closing soon, if you have any further questions or comments, please kindly let us know, and we are happy to respond.

Paper9690 authors

---

### Meta-Review · Area_Chair_uvJw · 2022-08-29

**Recommendation:** Accept
**Confidence:** Certain

**Metareview:**

This paper introduced a new matching mechanism in mask-level instead of pixel-level matching in previous few-shot segmentation methods. Additionally, they propose a feature-alignment block based on the attention mechanism to align both support and query features individually and cross-align between them. Reviewers in general agree on the novelty of the proposed approach. Some reviewers were concerned about some details of module design, and the authors have answered those satisfactorily. Finally, AC believes that this approach of producing mask proposals and matching at the mask level is a novel direction for few-shot segmentation, and recommends acceptance of the paper.

**Award:**

No

---

### Decision · Program_Chairs · 2022-09-14

Accept